# Association of HSF1 Genetic Variation with Heat Tolerance in Chinese Cattle

**DOI:** 10.3390/ani9121027

**Published:** 2019-11-25

**Authors:** Yu Rong, Mingfei Zeng, Xiwen Guan, Kaixing Qu, Jianyong Liu, Jicai Zhang, Hong Chen, Bizhi Huang, Chuzhao Lei

**Affiliations:** 1Key Laboratory of Animal Genetics, Breeding and Reproduction of Shaanxi Province, College of Animal Science and Technology, Northwest A & F University, Yangling 712100, China; rongyu97@126.com (Y.R.); manfay@nwafu.edu.cn (M.Z.); guanxiwen97@163.com (X.G.); chenhong1212@263.net (H.C.); 2Yunnan Academy of Grassland and Animal Science, Kunming 650212, China; kaixqu@163.com (K.Q.); ljy@ynbp.cn (J.L.); ynzjc@126.com (J.Z.)

**Keywords:** cattle, *HSF1* gene, SNP, heat tolerance, heat stress

## Abstract

**Simple Summary:**

China harbors two lineages of cattle (*Bos taurus* and *Bos indicus*). North China bears *Bos taurus*. *Bos indicus* predominantly belong to south China, whereas in central China there is an admixture of *Bos taurus* and *Bos indicus*, representing a discontinuity with north and south China. It has been reported that *Bos indicus* are more heat resistant than *Bos taurus* and heat stress can damage the structure of protein. Heat shock factor1 (HSF1) is a candidate heat tolerance gene which works by increasing expression of nascent heat shock proteins (HSPs), promoting the refolding of denatured proteins. Conversely, despite its importance, the effects of the bovine *HSF1* gene on heat tolerance traits in Chinese cattle remain unknown. Our results suggest that a missense mutation of the *HSF1* gene is associated with the heat tolerance capability of Chinese cattle.

**Abstract:**

The heat shock factor 1 (*HSF1*) gene is a regulator of the heat stress response, maximizing HSP protein expression survival. In this research, we explored the frequency distribution of a missense mutation (NC_037341.1 g.616087A > G, rs135258919) in the *HSF1* gene in Chinese cattle with amino acid substitution, valine to alanine. This mutation could be related to the heat tolerance in *Bos indicus*. A total of 941 individuals representing 35 Chinese native cattle breeds, combining pure taurine (Angus) and indicine cattle, were used to determine the genotypes of the mutation through PCR and partial DNA sequencing. The results showed significant differences in allele frequencies and their genotypes amongst Chinese cattle from different regions. Allele G or indicine-specific allele frequency diminished from south to north China, while allele A (genotype AA) or the taurine-specific allele had a contrary pattern, which agreed with the distribution of taurine and indicine cattle. According to the association analysis, the NC_037341.1 g.616087A > G (rs135258919) of the bovine *HSF1* gene, annual temperature (T), relative humidity (RH), and the temperature humidity index (THI) (*p* < 0.01) were interrelated closely, which indicated that the NC_037341.1 g.616087A > G of the *HSF1* gene is associated with heat tolerance in indicine cattle.

## 1. Introduction

Heat stress has some negative effects on cattle [1], such as decrease in production [2], reproduction [3], and immunity [4]. However, as global warming continues to increase [5,6,7], total economic losses due to animals suffering from heat stress, reached $2.4 billion annually in the United States [8]. Hence, mitigating the effect of heat stress is an important task for animals.

The *HSF1* gene is highly conserved in eukaryotes and is the main medium of protein toxic stress transcription response, playing a crucial role in non-stress regulation such as development and metabolism [9]. In *Caenorhabditis elegans,* overexpression of the *HSF1* gene can protect against heat stress compared with those with lower activity [10,11]. Consequently, the *HSF1* gene was considered to regulate the heat shock [12]. Interestingly, expression of HSF1 has been reported to be significantly higher in cattle during summers (*p* < 0.01), more so than any other season, thus associating its relationship with heat stress [13]. Moreover, a novel single nucleotide polymorphism(SNP) of the *HSF1* gene located in 3′UTR (G4693T) is considered to be associated with heat tolerance. The result from real-time PCR of locus 4693 suggested that the TT genotype can influence expression of the bovine *HSF1* gene which can enhance the heat tolerance [14]. In addition, some synonymous mutations were also related to heat tolerance [15]. It has been confirmed that heat stress can reduce the lactose of milk, which has been utilized for reactions other than milk synthesis in heat stress [16]. Besides, heat stress is associated with reduction in milk lactose, and a previous study represented the association of the *HSF1* gene (NC_037341.1 g. 616087A > G, rs135258919) with higher milk lactose content [17]. Furthermore, the distribution of the *HSF1* gene (NC_037341.1: g. 616087A > G, rs135258919) was also consistent with the distribution of *Bos taurus* and *Bos indicus* (http://animal.nwsuaf.edu.cn/code/index.php/BosVar) [18]. In support of the above literature, the potential SNPs in HSF1 may regulate heat tolerance; however, no missense mutation has been reported in the coding region of HSF1. In this study, we explored how a missense mutation in the *HSF1* gene (NC_037341.1 g. 616087A>G, rs135258919) is associated with heat tolerance in Chinese cattle.

China has abundant genetic resources for domestic cattle [19,20]. According to their geographical distribution and morphological characteristics, Chinese cattle breeds are composed of *Bos taurus* and *Bos indicus* [21]. The frequency of Chinese indicine cattle progressively decreases from south to north [22]. Additionally, China has tropical, subtropical, temperate, and cold regions that provide a geographic basis for SNPs variation [23], which make Chinese cattle suitable for the selection of heat-tolerant SNPs.

## 2. Materials and Methods 

### 2.1. Ethics Statement

The protocols used for the sampled animals in the current study were recognized by the Faculty Animal Policy and Welfare Committee of Northwest A & F University (FAPWC-NWAFU, Protocol number, NWAFAC1008).

### 2.2. Animal Source, DNA Preparation, and Data Collection

In this study, the genomic DNA of 35 native breeds (*n* = 941), Angus (*n* = 21) and Zebu (*n* = 60) (Appendix A) were extracted from the ear tissue or blood using the phenol–chloroform method [24]. Three environment parameters (temperature (T), relative humidity (RH) and temperature humidity index (THI)) for the 35 Chinese indigenous cattle breeds were collected from the Chinese Central Meteorological Office (http://data.cma.cn) over the last 30 years (1950–1980) (Appendix A). As suggested by McDowell et al., these parameters could be important indicators for the effects of heat and humidity [25]. T is degree Celsius and RH is relative humidity. The formulation of the THI (1.8T + 32) − (0.55-0.0055 RH) (1.8T–26) was based on the National Oceanic and Atmospheric Administration (1976), which was used to estimate thermal tolerance traits. Here, *Bos indicus* were used as the positive control due to extreme environments, such as the high temperature and the high burden of parasites in African and Asian regions [26,27], whereas Angus was regarded as the negative control due to its lesser ability to regulate heat stress, when compared to *Bos indicus*. *Bos indicus* carried more of the genes associated with heat tolerance [28,29]. The genotype frequency was based on DNA sequencing of the local breeds that were used, in addition to Zebu and Angus, as reference groups (Appendix A).

### 2.3. Primers Design, PCR Amplification and DNA Sequencing

The primers were designed for HSF1 (primers 5′-AGCAAATCCCATGCCGAGG and 5′-CGACCACCCTTATTGACTCCA). The final master mix of 12.5 µL PCR amplification composed of 10 ng of genomic DNA, 0.2 µL for each primer (20 pmol), 0.1 Mm dNTPs, 4.82 mL free water, 1x PCR buffer (2.5 mM of Mg^2+^) and 0.5 U rTaq DNA polymerase. The product of PCR (257 bp) was inspected by 1.0% Dured stained gel electrophoresis. DNA sequencing was detected using an ABI PRIZM 377 DNA sequencer by Sangon Biotech Company, Shanghai, China.

### 2.4. Data Analysis

SPSS software (Version 20.0, Inc, Armonk, New York, NY, USA) was used to correlate the value of the three environmental parameters (T, RH, THI) from the sampling sites. The formula of the model was a linear model (SPSS 18.0):Y_i_ = u+ G_i_ +B_i_ + e_i_
where Y_i_ =the value of T, RH, and THI from 1951 to 1980; u is the overall mean value; G_i_ = the fixed genotype effect; B_i_ = the fixed effect of breeds and e_i_ = the random residual effect. Differences were considered significant at *p* < 0.05.

The Hardy–Weinberg equilibrium (HWE) was tested based on a likelihood ratio for different locus–population combinations and the number of observed and effective alleles, using POPGENE software (Version 3.2) [30]. Population genetic indexes, such as the numbers of effective alleles (Ne), gene homozygosity (Ho), gene heterozygosity (He) (Ho + He = 1), were calculated by POPGENE software; He and Ho were calculated according to Nei’s (1973) method and were indicators for genetic variation [31]. Polymorphism information content (PIC) was calculated, according to Botstein et al’s (1980) methods [32].

## 3. Results

### 3.1. Distribution of Genotypic and Allelic Frequencies

The missense mutation of the *HSF1* gene (NC_037341.1 g.616087A > G, rs135258919) led to the transformation from valine into alanine (http://animal.nwsuaf.edu.cn/code/index.php/BosVar) [18]. From 941 individuals, three genotypes were found (AA, AG, GG), the frequencies of which were 0.2030, 0.3379, 0.4591, respectively. There was also a discontinuous distribution of allele G frequency in *Bos taurus* and *Bos indicus*. Allele G in southern, central and northern Chinese cattle reached 0.7727, 0.4692, and 0.1282, respectively, but allele A had a contrary pattern which enhances successively from south to north. Interestingly, the frequency of allele G was nonexistent in Angus and more than 0.8600 in Zebu (Appendix A).

### 3.2. Diversity Analysis

Ho, He, Ne, and PIC are showed in Appendix A, as genetic parameters of cattle. From the results, the maximum and minimum of He values were 0.06 and 0.50. Ne values were from 1.07 to 2.00. The PIC ranges were from 0.07 to 0.38. The PIC value showed that 15 indigenous cattle were at low genetic diversity, while the others showed moderate polymorphisms in the missense mutation (Appendix A). The χ2 test indicated that some southern Chinese cattle, such as BH (Bohai), and some south-western cattle, such as WS (Wenshan), DQ (Diqing), and SH (Shigatse Humped) were not as consistent with the Hardy–Weinberg equilibrium (*p* < 0.05). The Ne of six Chinese cattle breeds was >1.900.

### 3.3. Correlation Analysis of the HSF1 Gene for Heat Tolerance

The results of the correlation analysis between the three environment parameters (T, RH, and THI), collected from sample regions and genotypes, are presented in Table 1. It was assumed that there are very significant differences (*p* < 0.01) in the genotypes AA, AG, and GG. (Table 1). More individuals carrying the GG or AG genotype were detected in regions with more prominent heat stress than those with the AA genotype, which suggested that allele G and heat tolerance in Chinese cattle might be interrelated (Figure 1 and Appendix A). From the subject effect test on the *HSF1* genotypes, it was observed that RH and genotypes are interrelated very strongly (Appendix A).

## 4. Discussions

Heat stress can raise the body temperature, resulting in decline in feed intake and ultimately leading towards reduced productivity [2]. It reduces the duration and intensity of estrus, alters follicular development, and impairs the embryonic development of cattle [3]. Additionally, it decreases calf birth during the dry period of dairy cows and compromises the passive IgG transfer from colostrum and the cell-mediated immune function of the calves during the preweaning period [4]. Consequently, heat stress represents a major environmental factor that negatively affects the health and performance of dairy cows, causing huge economic losses to the dairy industry. Identifying and selecting animals that are thermo-tolerant is an attractive alternative for reducing the negative effects of heat stress on dairy cattle performance [33].

The *HSF1* gene is associated with heat stress regulation and maximizes heat shock protein (HSP) expression by synergizing with the *DAF-16* gene. HSP was considered as an intra-cellular chaperone, frequently involved in inflammation, endoplasmic reticulum stress, apoptosis at the liver and endothelial level, assisting protein folding, targeting damaged protein degradation, and other reactions to resist stress [34,35,36]. It was proved that overexpression of the *HSF1* gene can also inhibit cell apoptosis under heat stress, increasing the rate of cell survival so as to attenuate the effects of heat stress. Therefore, the *HSF1* gene is a sufficient and necessary regulator of heat tolerance [12]. As a result, the *HSF1* gene may potentially be the candidate gene responsible for heat tolerance.

There are contrary environments in China; it is dry and cold in north China while the climate in central China is propitious to agriculture. As for south China, it is humid and hot. *Bos indicus* are more heat-tolerant than *Bos taurus* [28]. There are mainly *Bos taurus* in north China while cattle in south China are predominantly *Bos indicus*. Both *Bos taurus* and *Bos indicus* are prevalent in central Chinese cattle through commercial activity and migration of the population [21,37,38,39].

In our study, we explored SNPs in the *HSF1* gene in the database of the Bovine Genome Variation Database and Selective Signatures (BGVD) (http://animal.nwsuaf.edu.cn/code/index.php/ BosVar), to confirm whether thermal tolerance is associated with polymorphisms in the *HSF1* gene [18]. A missense mutation (NC_037341.1 g. 616087A > G, rs135258919) was highly represented in *B. indicus* breeds but was absent in *B. taurus*. It is generally believed that due to evolutionary differences and adaptation to tropical conditions, *B. indicus* cattle have superior heat tolerant ability to *Bos taurus* cattle [40]. Intriguingly, Chinese cattle also have the same pattern. Genotypic diversity is important for species preservation and for the performance ofc selected animals. It was showed that south- western cattle such as TB (Tibetan), SH (Shigatse Humped), DQ (Diqing), WS (Wenshan) universally showed the high genetic diversity caused by the introgression of yak, *B. javanicus* [22]. *B. javanicus* is well adapted to the tropical climate environment, which may enhance the heat tolerance of south-western Chinese cattle (Figure 1).

The missense mutation (NC_037341.1: g. 616087A > G, rs135258919) was detected in the *HSF1* gene. Results represent that the frequency of allele G diminishes from south to north, indicating a significant difference in geography across indigenous Chinese cattle breeds and in agreement with the distribution of taurine and indicine cattle in China. As for the southern group, the highest frequencies of allele G were detected in south-eastern cattle. The south-east had higher environmental parameters than any other region followed by the south-west. Also, the lineage of southern Chinese cattle was mainly indicine [41,42,43,44] in areas whose values of T, RH, and THI were significantly (*p* < 0.01) higher than those that had genotype AA. Based on these findings, the mutation for the *HSF1* gene was associated with heat tolerance, which might be useful for marker-assisted selection of cattle.

## 5. Conclusions

In conclusion, our results indicated that mutation of the *HSF1* gene is associated with heat tolerance in Chinese indigenous cattle and might be a candidate gene for marker-assisted selection to improve the heat tolerance trait in Chinese indigenous cattle.

## Figures and Tables

**Figure 1 animals-09-01027-f001:**
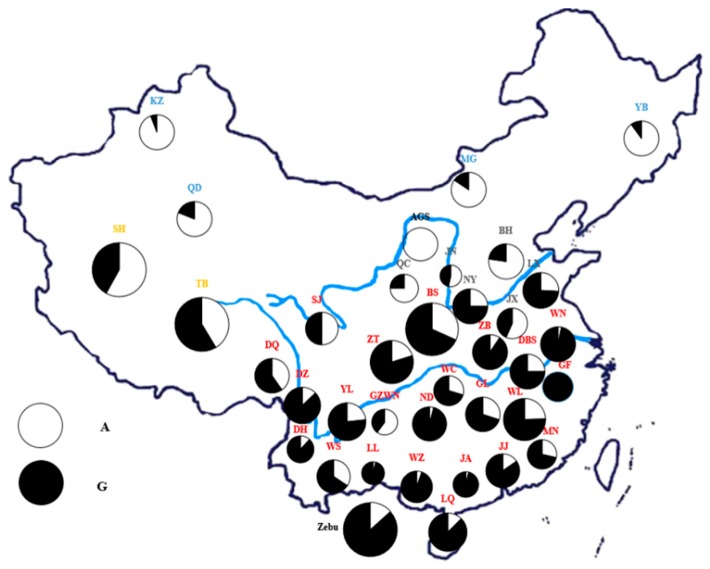
Geographical distribution of two variants among 35 Chinese breeds as well as the Augus and Zebu population. Distribution of G and A alleles of the NC_037341.1 g.616087A > G locus of the *HSF1* gene. BH, Bohai Black; BS, Bashan; DBS, Dabieshan; DH, Dehong; DQ, Diqing; DZ, Dianzhong; GF, Guangfeng; GL, Guanling; GZWN, Weining; JA, Ji’an; JJ, Jinjiang; JN, Jinnan; JX, Jiaxian red; KZ, Kazakh; LL, Longlin; LQ, Leiqiong; LX, Luxi; MG, Mongolian; MN, Minnan; ND, Nandan; NY, Nanyang; QC, Qinchuan; QD, Qaidam; SJ, Sanjiang; SH, Shigatse Humped; TB, Tibetan; WC, Wuchuan; WL, Wuling; WN, Wannan; WS, Wenshan; WZ, Weizhou; YB, Yanbian; YL, Yunling; ZB, Zaobei; ZT, Zhaotong (the different size means different scale of sample numbers).

**Table 1 animals-09-01027-t001:** Associations of polymorphisms in the *HSF1* gene, with the mean annual temperature, humidity, and temperature–humidity index in Chinese indigenous cattle.

Polymorphism	Genotype (n)	Temperature (°C) (LSM ± SE)	Relative Humidity (%) (LSM±SE)	Temperature–Humidity Index (LSM ± SE)
HSF-1 gene NC_037341.1 g.616087A > G, rs135258919	AA (191)	8.50 ^A^ ± 4.57	60.78 ^A^ ± 12.41	50.50 ^A^ ± 5.75
	AG (318)	12.60 ^B^ ± 5.13	70.22 ^B^ ± 10.11	55.52 ^B^ ± 7.30
	GG (432)	15.97 ^C^ ± 5.02	74.80 ^C^ ± 7.90	59.91 ^C^ ± 6.95

LSM ± SE, least squares means and their standard errors for each genotypic class reported. Uppercase letters mean differences of the value at *p < 0.01*.

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
