# Peer review of "Association of HSF1 Genetic Variation with Heat Tolerance in Chinese Cattle"

_animals, 2019, doi:10.3390/ani9121027_

Round 1
Reviewer 1 Report
The work is quite interesting and original. The experimental design should be better explained and for the amount of reported data it would be more useful a presentation via short communication. the english style must absolutely be reviewed. I recommend to contact an English native. The Discussion, as well as the Conclusions sections must be extensively revised and enriched. More information are needed in order to better understand the novelty of the study and its robustness
Minor Concerns:
L14: Replace as follow: “There were two bovine leneages in China…”
L15: Replace “blong” by “belong”
L19: “proteins(HSPs)” insert space
L43: explain what are the negative effects on immunity and reproduction
L49: Caenorhabditis elegans. Write in italic
L81: Check for double space
L82-84: “Three environment parameters (temperature, relative humidity, temperature humidity index) 1950-1980 were collected shown on Table S1” The sentence is not clear, please rewrite.
L105: Why did you consider significant only differences with P<0.01? I think it's too stringent
L138: Insert space: (Figure 1&
Author Response
Thanks for your comments, we have contacted an English native to polish my manuscript. We have enriched our discussion and marked red to revirew for convenience. We also repalce the inappropriate formulation such as the L14:“There were two bovine leneages in China…. Some wrong with spelling such as L19 and, double space(L81) and the missing space(L138) were checked. As for the negative impact of heat stress in reproduction and immunity. Heat stress can rise the body temperature, resulting in decline in the feed intake, ultimately leading towards reduced productivity [2]. It reduces duration and intensity of estrus, altered follicular development, and impaired embryonic development of cattle under heat stress [3]. Besides, it also decreases calf birth during the dry period of dairy cows and compromised the passive IgG transfer from colostrum and cell-mediated immune function of the calves during the preweaning period [4]. And we have added them in the discussion(L147-152). Then we have rephrased the L82-84. Three environment parameters (temperature (T), relative humidity (RH), temperature humidity index (THI)) for the 35 Chinese indigenous cattle breed were collected from the Chinese Central Meteorological Office (http://da ta.cma.cn) for over the last 30 years (1950 - 1980) (Table S1). The considering of significant differences have changed to 0.05. It is too stringent indeed.
Reviewer 2 Report
This study showed that mutation of the HSF1 gene is associated with heat tolerance in Chinese indigenous cattle, which might be a candidate gene for the marker-assisted selection to improve the heat tolerance trait in Chinese indigenous cattle.
Heat shock proteins (HSPs) function as intra-cellular chaperones, frequently involved in inflammation, endoplasmic reticulum stress and apoptosis at liver and endothelial level [1]. Heat stress represents a major environmental factor that negatively affects the health and performance of dairy cows, causing huge economic losses to the dairy industry [2]. Identifying and selecting animals that are thermotolerant is an attractive alternative for reducing the negative effects of heat stress on dairy cattle performance [2]. It is generally believed that due to evolutionary differences and adaptation to tropical conditions, Indian native cattle has superior heat tolerant ability than Bos taurus cattle [3].
Authors are kindly requested to emphasize the current concepts about these issues in the context of recent knowledge and the available literature. This articles should be quoted in the References list. In addition, figures and tables should be made readable.
References
Carotid intima-media thickness is predicted by combined eotaxin levels and severity of hepatic steatosis at ultrasonography in obese patients with Nonalcoholic Fatty Liver Disease. PLoS One. 2014 Sep 30;9(9):e105610. doi:10.1371/journal.pone.0105610. Whole Genome Mapping Reveals Novel Genes and Pathways Involved in Milk Production Under Heat Stress in US Holstein Cows. Front Genet. 2019 Oct 4;10:928. doi:10.3389/fgene.2019.00928. Characterizing binding sites of heat responsive microRNAs and their expression pattern in heat stressed PBMCs of native cattle, exotic cattle and riverine buffaloes. Mol Biol Rep. 2019 Oct 22. doi: 10.1007/s11033-019-05097-8.Author Response
1.Thanks for your comments, we have replaced some references. And we also added these references in our manuscript. But we added them in our discussion to explain the negative impacts of heat stress and the importance of heat tolerance out the length of the introduction and the discussion.
2. We also rephrased the articulation of HSPs.

Round 2
Reviewer 1 Report
Authors correctly addressed all my suggestions and requests for clarification.